# Anatomically Standardized Detection of MRI Atrophy Patterns in Early-Stage Alzheimer’s Disease

**DOI:** 10.3390/brainsci11111491

**Published:** 2021-11-11

**Authors:** Lukas Lenhart, Stephan Seiler, Lukas Pirpamer, Georg Goebel, Thomas Potrusil, Michaela Wagner, Peter Dal Bianco, Gerhard Ransmayr, Reinhold Schmidt, Thomas Benke, Christoph Scherfler

**Affiliations:** 1Department of Neurology, Medical University of Innsbruck, 6020 Innsbruck, Austria; lukas.lenhart@i-med.ac.at (L.L.); thomas.potrusil@gmx.at (T.P.); thomas.benke@i-med.ac.at (T.B.); 2Department of Neuroradiology, Medical University of Innsbruck, 6020 Innsbruck, Austria; michaela.wagner@tirol-kliniken.at; 3Center for Neurosciences, Department of Neurology, University of California, Davis, CA 95616, USA; sseiler@ucdavis.edu; 4Imaging of Dementia and Aging (IDeA) Laboratory, Davis, CA 95616, USA; 5Department of Neurology, Medical University of Graz, 8036 Graz, Austria; lukas.pirpamer@medunigraz.at (L.P.); reinhold.schmidt@medunigraz.at (R.S.); 6Department of Medical Statistics, Informatics and Health Economics, Medical University of Innsbruck, Müllerstraße 44, 6020 Innsbruck, Austria; georg.goebel@i-med.ac.at; 7Department of Neurology, Medical University of Vienna, 1090 Vienna, Austria; peter@dal-bianco.at; 8Department of Neurology, Kepler University Hospital, 4021 Linz, Austria; Gerhard.Ransmayr@kepleruniklinikum.at

**Keywords:** Alzheimer’s disease, structural magnetic resonance imaging, cortical thickness, hippocampal subfields

## Abstract

MRI studies have consistently identified atrophy patterns in Alzheimer’s disease (AD) through a whole-brain voxel-based analysis, but efforts to investigate morphometric profiles using anatomically standardized and automated whole-brain ROI analyses, performed at the individual subject space, are still lacking. In this study we aimed (i) to utilize atlas-derived measurements of cortical thickness and subcortical volumes, including of the hippocampal subfields, to identify atrophy patterns in early-stage AD, and (ii) to compare cognitive profiles at baseline and during a one-year follow-up of those previously identified morphometric AD subtypes to predict disease progression. Through a prospectively recruited multi-center study, conducted at four Austrian sites, 120 patients were included with probable AD, a disease onset beyond 60 years and a clinical dementia rating of ≤1. Morphometric measures of T1-weighted images were obtained using FreeSurfer. A principal component and subsequent cluster analysis identified four morphometric subtypes, including (i) hippocampal predominant (30.8%), (ii) hippocampal-temporo-parietal (29.2%), (iii) parieto-temporal (hippocampal sparing, 20.8%) and (iv) hippocampal-temporal (19.2%) atrophy patterns that were associated with phenotypes differing predominately in the presentation and progression of verbal memory and visuospatial impairments. These morphologically distinct subtypes are based on standardized brain regions, which are anatomically defined and freely accessible so as to validate its diagnostic accuracy and enhance the prediction of disease progression.

## 1. Introduction

Alzheimer’s disease (AD) is a progressive neurodegenerative disorder with the insidious onset of memory decline and the subsequent impairment of multiple cognitive domains. Neuropathologically, the disease is characterized by deposits of phosphorylated tau protein (neurofibrillary tangles, NFT) and the aggregation of beta amyloid (Aβ) peptides (amyloid plaques). AD typically originates in the medial temporal lobes, which include the hippocampus, and spreads to the association cortices [1]. Recent postmortem examinations revealed that 25% of AD cases did not follow Braak staging, indicating distinct AD subtypes. Three subtypes were recently described and termed as (i) typical AD that has generally balanced NFT counts in the hippocampus and the medial temporal lobe, also including the association cortices, respectively, (ii) limbic-predominant AD presenting with NFT accumulations predominantly in the hippocampus, and (iii) hippocampal-sparing AD showing NFT counts predominantly in the association cortex [2]. NFT distribution patterns of AD subtypes have been further shown to considerably overlap with AD atrophy patterns, as revealed using volumetric MRI [3,4,5]. Individuals with the hippocampal-sparing subtype of AD were associated with a younger age at onset, shorter disease duration, rapid progression and the presence of a non-amnestic syndrome compared to those with typical AD. In contrast, limbic-predominant AD was typically associated with late disease onset, slower disease progression and an amnestic syndrome [2,5,6,7,8].

Voxel-based and surface-based morphometry measures are commonly used to accurately localize the patterns of brain atrophy [9,10,11]. Through the application of unsupervised classification methods to automated MRI analysis techniques, morphometric AD subtypes were successfully identified in an unbiased fashion [7,8,12,13,14,15,16]. Additionally, a considerable overlap of AD atrophy patterns was identified via a voxel-based analysis, confirmed by the distribution of neurofibrillary tangle counts at autopsy [5]. Although these types of MRI post-processing approaches have the potential to reveal disease related signal alterations on the group level, efforts to provide disease specific markers for the clinical routine work-up are limited due to a lack of the free availability of such identified voxel clusters, as well as due to difficulties relating to their standardization and implementation into diagnostic algorithms. These concerns can be managed by performing an automated compartmentalization of the entire brain using predefined and atlas-conforming regions of interest [17]. The open-source MRI post-processing algorithm “FreeSurfer” meets these requirements and further permits the calculation of morphometric parameters such as the cortical thicknesses and subcortical volumes [10,11]. The quantification of regionally and individually measured cortical thicknesses outperformed the volumetric assessments regarding inverse correlations with Braak staging at pathology and was therefore the prioritized methodology for cortical analysis over the estimation of regional brain volume estimates [18]. Morphometric parameters such as the hippocampal volume and the entorhinal and supramarginal thicknesses were found to be useful as non-invasive surrogate markers for mild cognitive impairment and AD dementia [19]. Recently, the FreeSurfer software package was extended to provide a more precise segmentation of the hippocampus into several interconnected and functionally different hippocampal subfields [20]. By standardizing the MRI parameters using a large age- and sex-matched healthy control data set through z-transformation, the morphometric measures can be ranked according to their severity of atrophy on the single subject level, which in turn allows for the distinction between ‘signal distribution’ and ‘signal quantity’ to be made.

The objectives of this exploratory and data-driven study were (i) to apply atlas-derived freely accessible individual measurements of cortical thickness and subcortical volumes, including the hippocampal subfields, so as to identify distinct atrophy patterns on the group level of a newly and prospectively recruited multicenter cohort of patients with early stage probable AD, and (ii) to compare the demographic and cognitive profiles at baseline and for a one-year follow-up of those previously identified morphometric AD subtypes in order to predict disease progression.

## 2. Materials and Methods

### 2.1. Study Population

Patients were identified from the prospective registry on dementia (PRODEM), an Austrian longitudinal multi-center cohort study conducted at four university hospitals between 2009 and 2016. In order to verify their eligibility, participants were required to satisfy the following inclusion criteria: (1) a diagnosis of probable Alzheimer-dementia according the NIA-AA criteria [21], (2) the availability of a caregiver who agreed to provide information on the patients’ condition, (3) non-institutionalization and without a need for 24-h care, (4) a the performing of a T1-weighted 3D MRI of the whole brain according to a standardized protocol and (5) the provision of available informed consent, signed by the patient. Detailed interviews at the time of the neurological examinations were conducted by trained behavioral neurologists and neuropsychologists. In order to (i) exclude patients with early onset AD and (ii) identify patients at early disease stages, we further limited the inclusion of patients to an age of an onset of above 60 years and a disease severity of less or equal to 1, assessed via the clinical dementia rating scale (CDR) [22]. Subjects with a history of cerebral infarctions, hemorrhage, tumors, hydrocephalus, or severe head trauma were excluded from participation. We also excluded patients with MRI scans showing confluent white matter lesions according to the Fazekas score 3 (*n* = 15) [23] to (i) keep the inclusion of patients with vascular dementia at a minimum and (ii) to avoid jeopardizing the automated segmentation of adjacent subcortical brain volumes. Furthermore, the segmentation of the MRI failed in *n* = 5 patients. The final study population consisted of 120 patients aged between 60 and 82.5 years. Of the 120 patients, 74 (62%) had a 3T MRI and 46 patients had a 1.5T MRI. The control cohort comprised of a total of 348 healthy individuals (*n* = 214 females, 61.5%). Of the 348 participants, 242 (69%) had a 3T MRI and were recruited by the Austrian Stroke Prevention-Family-Study (*n* = 242) [24]. The remaining 106 participants underwent a 1.5T MRI and were resourced using the local inhouse databases of the study MRI scanners. The entire healthy control cohort was characterized by normal cognitive functions, determined by neuropsychological tests and none of the participants had a history of neurological or psychiatric disorders. The age of the control group ranged from 60 to 82.5 years (72.52 ± 6.47 years). Among this healthy control group, a sex-matched cohort of at least 44 subjects with an age range of ±5 years was selected to serve as the healthy subjects’ sample to enable the z-transformation of regional morphometric measures for each patient. PRODEM was approved by the Ethic Committees of the Medical Universities of Vienna (Vote from 30.06.2008, Code 176/2008), Graz (Vote from 13.03.2008, Code 19-135 ex 07/08), and Innsbruck (Vote from 03.11.2009, Code UN3259, Conference Number 266/4.8), as well as the General Hospital of Linz (no number was assigned by the ethics committee of Linz). The patients’ and the caregivers’ written informed consent was obtained according to the Declaration of Helsinki.

### 2.2. Neuropsychological Tests

To conduct the assessment of cognitive functions, the standardized neuropsychological battery, the ‘‘Consortium to Establish a Registry for Alzheimer’s disease (CERAD)—Plus’’, was used [25]. To avoid the occurrence of intercorrelations between CERAD subtests we selected semantic verbal fluency, the short version of the Boston Naming Test [26], the Mini-Mental State Examination (MMSE) [27] as a measure of global cognition, word list recall of a 10-word list to test verbal memory, constructional praxis to test visuospatial functions and constructional praxis recall tapping figural memory as the core variables representing important domains of cognitive decline [28]. The test results were adjusted for age, sex, and years of formal education. The CDR was used to provide a global evaluation of the severity of dementia. The Disability Assessment for Dementia Scale (DAD) quantitatively measured their functional abilities for activities of daily living, including basic, instrumental, and leisure activities [29]. Behavioral and psychological symptoms of dementia were assessed using the Neuropsychiatric Inventory (NPI). The short form of the Geriatric Depression Scale (GDS) was used to identify depression in AD patients [30].

### 2.3. MRI Acquisition

The MRI scans of patients and the healthy control participants were acquired through three different Siemens scanners (Avanto, SymphonyTim and TrioTim) with field strengths of 1.5 and 3 Tesla, using a predefined standardized protocol. The MRI conventional protocol included a high-resolution T1-weighted 3D MPRAGE sequence in 1 mm isotropic resolution coverage (TR = 1900 ms, TE = 2.19 ms, TI = 1100/900 ms (1.5/3T), flip angle = 9, isotropic resolution = 1 mm) and an axial T2-weighted FLAIR sequence (TR = 10,000 ms, TE = 69 ms, TI = 2500 ms, TI = 800/1100 ms (1.5/3T), number of slices = 40, slice thickness = 3 mm, in-plane resolution = 0.9 × 0.9 mm^2^.

### 2.4. Data Preprocessing

A morphometric analysis comprising the estimation of cortical thickness, subcortical and hippocampal volumes, including the subfield analysis, was conducted using the FreeSurfer software package version 6.0 (http://surfer.-nmr.mgh.harvard.edu, accessed on 3 November 2021). The image processing stream included the removal of non-brain tissue using a hybrid watershed/surface deformation procedure, automated Talairach transformation, intensity normalization and segmentation of the subcortical white matter and deep gray matter volumetric structures, including the hippocampus [31]. Subsequently, a tessellation of the gray matter white matter boundary was performed, following automated topology correction [32] and surface deformation according to intensity gradients to optimally place the gray/white and gray/cerebrospinal fluid borders at the location at which the greatest shift in intensity defines a transition to the other tissue class [17,33,34]. This method uses both intensity and continuity information from the entire 3D MRI volume in segmentation and deformation procedures to produce representations of the cortical thickness, calculated as the closest distance from the gray/white boundary to the gray/CSF boundary at each vertex on the tessellated surface [10]. The hippocampus was further segmented into its hippocampal subfields using a Bayesian inference approach and a novel atlas algorithm of the hippocampal formations [20]. Only the subfields that include ROIs of gray matter signal, that are appropriately delineable on 3D MPRAGE MRI, such as the left and right parasubiculum, presubiculum, subiculum, cornu ammonis (CA) 1, CA2–3, CA4, and the hippocampal tail were used for the analysis. The preprocessing steps were visually inspected to ensure that a misalignment of brain structures had not occurred. All imaging data were processed on a HP DL360p server with a total of 48 CPUs. For the statistical analysis, raw MRI volumetric measures were adjusted by calculating the ratio between the corresponding regional volume of interest and the subject’s estimated total intracranial volume [35,36]. To obtain the bihemispheric presence of the brain region, the value of the side with the lower z-score was entered to the PCA.

### 2.5. Statistical Analysis

Demographic data are presented as frequencies (percentage), means (± standard deviation) or median (interquartile range) according to data distribution. Gaussian distribution was confirmed via a visual analysis of the Q-Q plots and the Kolmogorov-Smirnov test. The cross sectional baseline and follow-up group differences as well as the longitudinal differences within and between the group of normally distributed data were analyzed by a mixed effect generalized linear model.

Either a Kruskal-Wallis one-way ANOVA by ranks, or a Mann-Whitney U test was applied as appropriate for non-Gaussian distributed variables. The Benjamini and Hochberg false discovery rate (FDR) was used for the correction of multiple comparisons to maintain the false positive rate at 5%. To test the scanner-related differences of volume and cortical thickness, measurement between the two MRI control data sets and patient samples, general linear models were established, with each MRI parameter set as a dependent variable, the scanner type as factor and age and sex as covariates.

A principal component analysis (PCA) with orthogonal rotation (varimax) was conducted and included 93 variables of the whole brain cortical thickness and subcortical volumes. In short, the aim of the PCA was to condense the information contained in a data set of 76 cortical regions and the volume measures of 5 hippocampal subfields and 12 subcortical brain regions into a more manageable set of independent components [37]. The PCA accounts for correlations among variables and is commonly used for dimensionality reduction by transforming a large set of variables into lower-dimensional data that contains most of the information contained in the entire data set. As a result, we identified three principal components and arranged them by decreasing order (of their eigenvalues), so that the first principal component combines the maximum possible information from all initial variables of the data set followed by the second principal component containing the maximum remaining information and so on. We used the Kaiser-Meyer-Olkin measure (KMO) to verify the sampling adequacy for the analysis (KMO = 0.833). Bartlett’s test of sphericity indicated that the correlations between items were sufficient for PCA (Chi2 (91 degrees of freedom) = 1851.11, *p* < 0.001). Subsequently, an unsupervised hierarchical cluster analysis was performed using the factor regression scores of the three principal components. Ward’s clustering linkage method was applied to combine pairs of clusters at each step while minimizing the sum of square errors from the cluster mean [38]. Each of the 120 patients with AD dementia was placed in their own starting cluster and then progressively clustered with others. To define the number of relevant morphometric patterns, the dendrogram was cut at the highest change of the similarity index, which represents a measure of congruency of morphometric parameters among patients. This procedure revealed four morphometric patterns, consisting of the brain regions identified by the PCA (Figure 1). SPSS version 24 was used (SPSS Inc., Chicago, IL, USA) to perform a statistical analysis.

## 3. Results

There was no significant main effect of the scanner type (1.5T and 3T MRI) on the MRI parameters, including regional brain volumes and cortical thickness measurements within the control data sets and within the patient samples.

### 3.1. Principal Component Analysis

Three components had eigenvalues over Kaiser’s criterion of 1 and, in combination, they accounted for 80.52% of the variance. The first component highly loaded on the following MRI parameters: the volumes of the entire hippocampus and the hippocampal subfields including the subiculum, the CA3, the CA4/dentate gyrus and the CA1-2 transition zones (L = 0.43, Table 1). The second component consisted of the mean cortical thickness measures of the superior parietal lobule, the precuneus, the cingulate sulcus marginal branch, the intraparietal sulcus and transverse parietal sulci, as well as the postcentral sulcus (L = 0.24). The third component included the cortical thickness of the superior temporal gyrus planum polare, the temporal pole, the inferior segment of the circular sulcus of the insula and the anterior transverse collateral sulcus (L = 0.13). Omitting the segmentation of the hippocampal subfields would have reduced the explained variance by 13%.

### 3.2. Morphological AD Subtypes Identified by Cluster Analysis

The greatest distance of the horizontal axis of the dendrogram from the cluster analysis, representing the measure of dissimilarity between patients, revealed four morphometric patterns, consisting of the brain regions identified by the PCA (Figure 1). At the 4-cluster level, patients were assigned, according to their cortical and subcortical atrophy patterns, into the hippocampal predominant (*n* = 37; 30.8%), the hippocampal-temporo-parietal (*n* = 35; 29.2%), the parieto-temporal (*n* = 25; 20.8%) and the hippocampal-temporal predominant (*n* = 23; 19.2%) morphometric subtypes (Figure 2, Table 2). A volume loss of the hippocampus and hippocampal subfields was observed in the hippocampal predominant, the hippocampal-temporo-parietal and the hippocampal-temporal predominant subtypes. Conversely, the hippocampal structures were relatively preserved in the parieto-temporal subtype. Cortical gray matter atrophy of parietal regions was evident in patients of the parieto-temporal, the hippocampal-temporo-parietal, and the hippocampal-temporal predominant subtypes. Temporal gray matter atrophy was a core feature of the hippocampal-temporal predominant subtype but was also present in the in the hippocampal-temporo-parietal and the parieto-temporal subtypes.

### 3.3. Demographic and Clinical Characteristics among the 4 AD Subtypes

The demographics and clinical characteristics of the 4 AD dementia subtypes are presented in Table 2 for baseline and at one-year follow-up. The mixed effect generalized linear model revealed neither a significant cross-sectional at baseline and one-year follow-up between groups nor longitudinal within-group differences regarding age, age at onset, disease duration, the NPI, GDS or DAD. Sex distribution was equal among the subgroups as well as the disease severity, as assessed by CDR and MMSE.

### 3.4. Cognitive Characteristics among the Four AD Dementia Subtypes

#### 3.4.1. Cross-Sectional Group Comparisons

Significant differences in the cognitive tests among AD subtypes at baseline and one-year follow-up are presented in Figure 3 and Table 2. At baseline, significant group differences between the four AD subtypes emerged regarding verbal fluency, naming, verbal memory, visuospatial functions, the MMSE and the CERAD total score. After one-year, cognitive deficits had progressed in all subgroups and for most domains, including in relation to global cognition (MMSE) and the CERAD total score, and significant differences between groups became evident for the cognitive variables of verbal fluency, naming and the visuospatial functions. Overall, the hippocampal-temporal subtype had the most advanced and the hippocampal predominant subtype had the least advanced cognitive impairment pattern, both, at baseline and during follow-up. Verbal memory was impaired in all groups, but relatively preserved in the parieto-temporal subgroup. In contrast, visuospatial functions were most reduced in the parieto-temporal group and best preserved in the hippocampal predominant group. Naming was in the subnormal range of the hippocampal predominant and hippocampal temporo-parietal groups, and significantly impaired in the parieto-temporal and hippocampal-temporal groups. Verbal fluency was most deteriorated in the hippocampal-temporal subgroup, and relatively preserved in the hippocampal predominant group.

#### 3.4.2. Longitudinal Group Comparisons

Between the baseline and follow-up investigation, the hippocampal-temporal group showed significant relative decreases in naming and verbal fluency scores when compared to the hippocampal-temporo-parietal group and the hippocampal predominant group, respectively.

## 4. Discussion

By applying the standardized and reproducible measurements of cortical thickness and subcortical volumes, including hippocampal subfields, this explorative, data-driven multicenter MRI study revealed four different brain atrophy patterns in a newly established cohort of patients with early-stage probable Alzheimer’s disease, defined by a CDR score of 0.5 to 1 and for a disease onset beyond the age of 60 years. The application of a priori atlas-based regions allowed for a precise and reproducible delineation of the anatomically defined brain areas on the individual MRI. Their further combination with quantitative measurements adjusted for age and sex constitutes a unique feature of this study. Due to the limited availability of automated ROI-based MRI analyses in AD, we attempted to compare the results of our study with voxel-based approaches that, although delineated brain atrophy more precisely, must be mapped to coarsely defined brain regions a posterior. These methodological differences limit the ability to compare more detailed atrophy patterns among studies and leads to discussions on results that are based on broader compartments of the brain.

At the 4-cluster level, a hippocampal predominant and a combined hippocampal temporo-parietal pattern were identified as the two most frequent subtypes, followed by a parieto-temporal type in which the hippocampus was spared and a hippocampal-temporal predominant subtype. Hippocampal atrophy was a core feature, of three of the four AD subtypes, i.e., 79%, which is in line with the Braak staging scheme. Combined, the hippocampal-temporo-parietal and the hippocampal-temporal predominant subtypes comprised 50% of the entire cohort and their frequency corresponded with previously reported diffuse or more wide-spread manifestations of AD atrophy patterns that include the hippocampus [6,8,39,40]. We also encountered a considerable proportion of patients presenting with more focal atrophy of either the hippocampus (i.e., 30.8%) or brain regions of the parietal and temporal lobe without a marked involvement of the hippocampus (i.e., 20.8%). The morphometric profile of these two subtypes corresponds with the description of the limbic-predominant variant and the hippocampal sparing variant respectively [5,6,8,39]. Interestingly, the frequency of the hippocampal predominant subtype was around 1.5 times higher in our AD cohort. This discrepancy probably arises from differences in the patients’ characteristics, image-post processing, and classification methods. Previous studies frequently included AD patients at later and more severe disease stages, which is associated with a higher proportion of cases with progressive and hence more diffuse atrophy patterns. In this study, the initial MRI acquired at an early disease stage (i.e., CDR 0.5 to 1) was used and when segmenting the hippocampus resulting subfields such as the CA1, CA3, CA4 and the subiculum outperformed the estimates of the entire hippocampal volume by finding 13% more of the common variance. This finding corresponds with previous studies, reporting higher sensitivity and diagnostic accuracy rates of the hippocampal subfield volumetry compared with whole hippocampus, to identify subtypes with hippocampal involvement. This strongly suggests an evaluation of the standardized parcellation of the hippocampus to a further degree as a surrogate marker to determine its accuracy for the classification of single patient presenting with dementia [40,41,42,43]. Atrophy patterns of the temporo-parietal subtype predominately consisted of the superior parietal lobule, the precuneus as well as the intraparietal and transverse parietal sulci, which is in agreement with former voxel-based morphometry studies and neuropathological observations [5,44]. The frequency of patients grouped into the temporo-parietal subtype was similar to the frequency of those reported in previous studies [14,39].

Clinical characteristics including age, gender and disease severity as assessed by the CDR and DAD, did not significantly differ across AD subtypes at baseline and one-year follow-up. In contrast, the profile of cognitive impairment varied considerably between subtypes [13,45,46]. At the baseline assessment, overall cognitive functioning measured by the CERAD total score was more impaired in those subtypes with widespread cortical atrophy in contrast to the hippocampal-predominant subtype. This finding is in line with recent voxel-based analyses that identified more diffuse atrophy patterns affecting the frontal, parietal and temporal lobe that were also associated with greater overall cognitive deficits [6,7,8,16].

Based on the CERAD subtests, two contrasting core deficit patterns were evident in our AD cohort. Pattern 1 was characterized by predominant verbal memory impairments but relatively preserved visuospatial functions. This pattern was found in most of the patients classified to the hippocampal predominant and hippocampal-temporal subgroup. By contrast, pattern 2 involved mainly visuospatial deficits in contrast to a relative preservation of memory. Since demographical and disease related factors are highly unlikely to account for this discrepancy in our study cohort, we suggest that the observed cognitive clusters are the result of the individual lesion patterns [47]. Accordingly, verbal memory deficits were more pronounced in the atrophy subtypes including the hippocampal formation, and visuospatial impairments mainly followed parietal lesions. In addition, impairments of verbal fluency and naming were common in atrophy subtypes involving the temporal and parietal cortex. Further studies are required to assess whether neuroimaging and cognitive heterogeneities reflect different underlying pathological processes and, consequently, if they are useful for predicting the disease progression of AD [8,39,45].

This study has several limitations. The inclusion of data from different MRI sites introduces potential issues due to different magnetic field strengths and scanner specific parameter setups. To minimize scanner related variability all MRI acquisitions were performed according to a predefined MRI protocol and image postprocessing was conducted at a single center. The variability of combing 1.5 and 3 Tesla cross-field strengths for cortical thickness and subcortical volume measures was shown to be insignificant when MRI acquisition protocols and data processing algorithms were controlled [48]. In addition, the trans-platform reliability of 1.5 Tesla versus 3 Tesla MRI of hippocampal subfield segmentation using FreeSurfer version 6.0 was demonstrated to be accurate in all human hippocampal subregions, except for the hippocampal fissure, which was not included in this study [49]. We have also limited the hippocampal subfield analysis to the segments of the gray matter compartment including CA1, CA2/3, CA4, the subiculum and the parasubiculum due to the fact that only of T1-weighted images were available. Furthermore, according to inaccuracies in the computer driven delineation of the occipital pole as well as the superior and middle occipital gyrus from surrounding subarachnoid space, these regions were excluded from the PCA. Other limitations concern the lack of diagnostic confirmation due to biomarkers such as CSF and PET imaging. Although our patient sample corresponds to probable AD with increased level of certainty, due to repetitive clinical visits with no change in diagnosis, documented cognitive decline, and congruent MRI findings [21], an amyloid and tau positivity as claimed by Jack et al. [50] was not available in a considerable portion of study participants and, therefore, the presence of other forms of neurodegenerative dementia cannot be entirely excluded.

Our study also has several strengths. These include the inclusion of only probable AD cases and the application of a standardized neuropsychological test battery. Besides the multicenter design and the inclusion of cortical as well as subcortical brain regions, the application of both a principal component analysis and a subsequent cluster analysis of high-dimensional data allowed for the identification of fourteen key MRI parameters building four AD atrophy patterns in an unsupervised fashion. Those parameters were defined by the freely available FreeSurfer atlas and hence available for the computation and validation of diagnostic algorithms.

## 5. Conclusions

In agreement with large multicenter cohorts as the Alzheimer’s Disease Neuroimaging Initiative (ADNI) or European community founded AddNeuroMed consortium, this study revealed, in a newly and prospectively recruited cohort at four Austrian sites, that AD is an umbrella term for morphologically distinct subgroups that also differ as to their cognitive phenotype. Morphological variability is already present in early-stage AD and can be localized when measuring cortical thickness and volumes of hippocampal subfields by applying a standardized and freely accessible brain atlas. Although grouped data is reported, atlas-based morphometric MRI parameters, other than atrophy patterns defined by voxel-clusters, can be translated to the individual subject level when accounting for age and gender. In this respect, further studies are warranted to validate the diagnostic accuracy of our approach and to strengthen predictions of disease progression.

## Figures and Tables

**Figure 1 brainsci-11-01491-f001:**
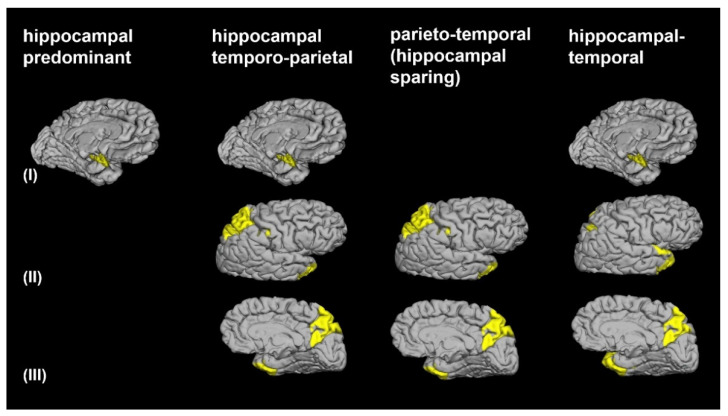
Morphometric patterns of gray matter atrophy. These subtypes were revealed at the four-cluster level, consisting of the brain regions identified by the principal component analysis, marked in yellow, such as (**I**) the hippocampus including the subiculum, the CA3, the CA4/dentate gyrus and the CA1-2 transition zones, (**II**) the superior parietal lobule, the precuneus, the cingulate sulcus marginal branch, the intraparietal sulcus and transverse parietal sulci, and the postcentral sulcus, (**III**) the superior temporal gyrus planum polare, the temporal pole, the inferior segment of the circular sulcus of the insula and the anterior transverse collateral sulcus. For cases with the bihemispheric presence of the brain region, the value of the side with the lower z-score is presented.

**Figure 2 brainsci-11-01491-f002:**
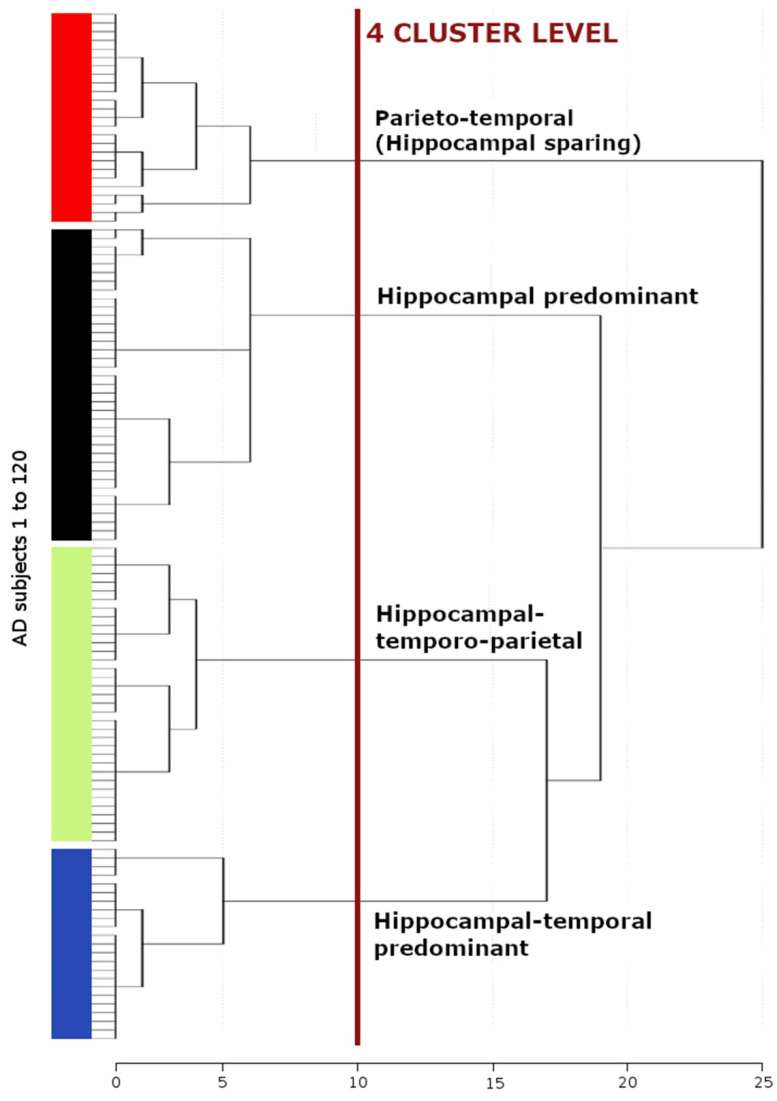
Hierarchical clustering dendrogram of morphometric Alzheimer’s disease subtypes. Ward’s linkage method was used to progressively cluster individual cases together. The distance along the *x*-axis represents the measure of similarity between patients, such that the longer the distance the greater the dissimilarity between the subjects, while the different colors of the vertical bar correspond to different Alzheimer’s disease subtypes. The red line reveals the clustered subtypes of AD dementia, which are illustrated in Figure 1.

**Figure 3 brainsci-11-01491-f003:**
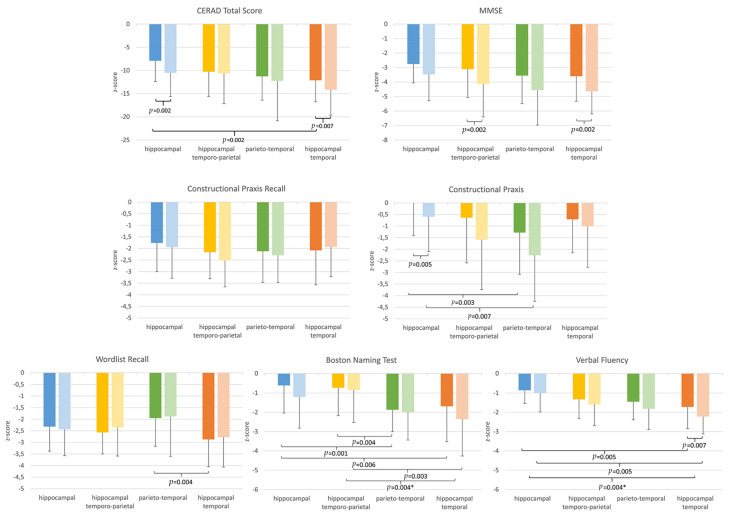
Neuropsychological test results. Bar plots of cognitive test scores of morphometrically defined Alzheimer’s disease subtypes at baseline (dark color) and one-year follow-up (light color). *p* values derived from group comparisons between AD subtypes. * Significant longitudinal differences of respective groups from baseline to follow-up time point.

**Table 1 brainsci-11-01491-t001:** Mean z-scores of brain regions identified by the principal component analysis.

	Hippocampal Predominant (*n* = 37)	Hippocampal-Temporo-Parietal (*n* = 35)	Parieto-Temporal (*n* = 25)	Hippocampal-Temporal Predominant (*n* = 23)
**Principal component I**				
Subiculum	−2.55 (1.32)	−3.50 (0.86)	−1.32 (0.87)	−3.41 (0.54)
CA1-2 transition zones	−2.40 (1.02)	−3.03 (0.83)	−0.96 (0.88)	−2.94 (0.65)
CA3	−2.24 (0.79)	−2.66 (0.63)	−0.35 (1.19)	−2.81 (0.82)
CA4/dentate gyrus	−2.71 (0.94)	−3.23 (0.76)	−0.72 (1.02)	−3.29 (0.84)
whole hippocampus	−2.80 (1.26)	−3.58 (0.83)	−1.21 (1.89)	−3.53 (0.60)
**Principal component II**				
superior parietal lobule	0.36 (1.36)	−2.06 (0.81)	−1.50 (1.67)	−1.22 (1.21)
Precuneus	0.26 (1.31)	−1.72 (1.08)	−1.59 (1.50)	−1.81 (0.98)
cingulate sulcus marginal branch	1.09 (1.38)	−1.01 (1.21)	−1.44 (1.06)	−0.82 (1.36)
intraparietal sulcus and transverse parietal sulci	0.06 (1.18)	−1.85 (0.87)	−2.15 (1.39)	−1.75 (0.96)
postcentral sulcus	0.56 (1.22)	−1.66 (0.82)	−1.55 (1.12)	−1.31 (0.80)
**Principal component III**				
superior temporal gyrus planum polare	−0.66 (1.55)	−1.22 (1.29)	−1.22 (1.35)	−3.79 (1.16)
temporal pole	−0.78 (1.57)	−2.12 (1.64)	−1.91 (1.76)	−4.37 (1.58)
inferior segment of the circular sulcus of the insula	−0.42 (1.38)	−0.34 (1.26)	−0.97 (1.42)	−2.86 (0.78)
anterior transverse collateral sulcus	−0.71 (1.36)	−1.05 (1.42)	−1.33 (1.62)	−3.03 (1.03)

Values represent the means of z-scores (±1 SD). The side of the regions with the lower value was selected as more affected side and included to the principal component analysis.

**Table 2 brainsci-11-01491-t002:** Demographic and clinical characteristics of patients with Alzheimer’s disease at baseline and one-year follow-up.

**At Baseline**	**Hippocampal Predominant**	**Hippocampal-Temporo-Parietal**	**Parieto-Temporal**	**Hippocampal-Temporal**	**Alzheimer’s Dementia**	**Control Group**
Sample size, *n* (%)	37 (38.8)	35 (29.2)	25 (20.8)	23 (19.2)	120	348
Age at MRI, years, mean (± 1 SD)	74.13 (4.92)	73.66 (4.8)	71.81 (5.38)	74.8 (5.44)	73.52 (5.43)	72.52 (6.47)
Sex, females, *n* (%)	25 (67.57)	20 (57.14)	17 (68)	15 (65.22)	77 (64.17)	210 (60.44)
Disease duration, years, median (IQR)	1.85 (2.35)	2.06 (2.65)	1.55 (2.32)	3.1 (2.41)	2.18 (2.55)	-
CDR, median (IQR)	0.5 (0.5)	0.5 (0.5)	0.5 (0.5)	0.5 (0.5)	0.5 (0.5)	-
GDS, median (IQR)	1 (2)	2 (2)	2 (2)	1 (2)	1.5 (2)	-
MMSE, mean (± 1 SD)	23.46 (3.17)	22.71 (4.59)	21.52 (4.16)	21.91 (4.40)	22.27 (4.3)	-
NPI, median (IQR)	9 (17)	4 (10)	4 (7)	2 (7)	4 (9)	-
DAD median %, (IQR)	85 (27.5)	92.5 (22.5)	84 (38)	87.5 (27.5)	88.49 (25)	-
**At One-Year Follow-Up**	**Hippocampal Predominant**	**Hippocampal-Temporo-Parietal**	**Parieto-Temporal**	**Hippocampal-Temporal**	**Alzheimer’s Dementia**	**Control Group**
Sample size, *n* (%)	24 (21.6)	30 (27)	16 (14.4)	20 (18)	90	-
Age at MRI, years, mean (± 1 SD)	74.73 (4.9)	74.21 (6.81)	74.2 (6.27)	75.84 (3.57)	74.46 (5.38)	-
Disease duration, years, median (IQR)	2.82 (2.98)	3.16 (2.6)	3.42 (1.84)	4.16 (3.3)	3.1 (2.48)	-
CDR, median (IQR)	1 (0.5)	1 (0.5)	1 (0.5)	1 (0.5)	1 (0.5)	-
GDS, median (IQR)	1 (2.75)	2 (3)	2 (2.5)	1 (3)	2 (2)	-
MMSE, mean (± 1 SD)	22.04 (3.86)	20.20 (6.00)	18.75 (5.47)	19.95 (4.01)	20.39 (4.9)	-
NPI, median (IQR)	9.5 (14.25)	8 (15.5)	5 (9.75)	2.5 (8.5)	6 (10.5)	-
DAD median %, (IQR)	65 (41.88)	67.5 (25.8)	70.83 (41.8)	75 (40)	71.25 (39.66)	-

CDR, clinical dementia rating scale; DAD, Disability Assessment for Dementia scale; GDS, geriatric depression scale; IQR, interquartile range; MMSE, Mini-Mental State Examination; NPI, neuropsychiatry inventory; SD, Standard deviation.

## Data Availability

The datasets used and analyzed for the present paper can be made available on request to the corresponding author due to privacy/ethical restrictions.

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
