# Peer review of "Anatomically Standardized Detection of MRI Atrophy Patterns in Early-Stage Alzheimer’s Disease"

_brainsci, 2021, doi:10.3390/brainsci11111491_

Round 1
Reviewer 1 Report
I am reviewing the manuscript entitled "Anatomically standardized detection of MRI atrophy patterns 2 in early-stage Alzheimer's disease."
The authors aim to find standardized and automated whole-brain MRI analysis, to increase the accuracy of the detection of Alzheimer's disease. The study purpose is highly valuable since objective measures are needed as biomarkers of this neurodegenerative disease.
Unsupervised MRI analysis has already proved to help differentiate AD subtypes. However, providing free available identified voxel clusters and the implementation of algorithms can further improve the diagnosis. So, the authors have used a free-access software, FreeSurfer, to facilitate the diagnosis. The objectives are clear and valuable for clinical practice.
The methods used for MRI analysis are clearly described. PCA was used to find the most relevant components.
However, I have some minor comments and questions for the authors:
- I find it difficult to understand that every component does not correspond to one single zone but a combination of several. Could you please provide a more extensive explanation?
- Table 1. Is there any relevance of positive values in the Principal component II? is it a mistake?
- Figure 1. The figure legend roman numbers are not indicated in the figure. The titles in each column should be more clearly separated (hippocampal-temporo-parietal and parieto-temporal titles are too close). Perhaps it would be easier to find the differences by aligning the similar atrophy zones (in the first line all and only the hippocampal images and so on).
- Figure 2. The dendrogram suggests possible sub-clusters in some cases. Was the clustering forced to give 4 clusters based on the assumption that this was the optimal number, or does the analysis suggest that? Please specify.
- There is no reference about the Control group in the Results about thickness nor in the cognitive tests. It is stated in the Methods section that they performed well in the cognitive test battery, but some reference in the statistics shown in Figure 3 is missing.
- Figure 3 is too small to be readable and becomes pixelated when zoomed in. Please correct it.
Author Response
1) I find it difficult to understand that every component does not correspond to one single zone but a combination of several. Could you please provide a more extensive explanation?
RESPONSE:
Thank you for this comment. Our data set consists of a considerable number of associated variables, such as volumes and cortical thickness values of adjacent brain regions, that might correlate with each other and hence might jeopardize statistical methods that assume independent parameters for the analysis. The number of variables and the possible associations among variables prompted us to use a principal component analysis, which was shown as appropriate approach in such kind of situations.
To improve the understanding of the reader, we modified and extended the paragraph about the description of the principal components analysis in the methods section on page 5 line 205-213 accordingly.
In short, the aim of PCA was to summarize the information contained in a data set of 76 cortical regions and volume measures of 5 hippocampal subfields and 12 subcortical brain regions into a more manageable set of independent components component factors explaining the maximal amount of common variance [37]. The PCA accounts for correlations among variables and is commonly used for dimensionality reduction by trans-forming a large set of variables into lower-dimensional data that still contains most of the information of the entire data collective. As a result, we identified three principal components and arranged them in decreasing order (of their eigenvalues), so that the first principal component combines the maximum possible information from all initial variables of the data set followed by the second principal component containing the maximum remaining information and so on.
2) Table 1. Is there any relevance of positive values in the Principal component II? is it a mistake?
RESPONSE:
The hippocampal predominant subtype was not affected by atrophy of morphometric parameters of principal component II that included the superior parietal lobule, the precuneus, the cingulate sulcus marginal branch, the intraparietal sulcus and transverse parietal sulci, and the postcentral sulcus. The slightly positive values represent relatively preserved cortical thickness in these regions of that subtype, which were not statistically significant compared to those of other subtypes.
3) Figure 1. The figure legend roman numbers are not indicated in the figure. The titles in each column should be more clearly separated (hippocampal-temporo-parietal and parieto-temporal titles are too close). Perhaps it would be easier to find the differences by aligning the similar atrophy zones (in the first line all and only the hippocampal images and so on).
RESPONSE:
We thank the reviewer for these suggestions, and we have modified Figure 1 accordingly.
4) Figure 2. The dendrogram suggests possible sub-clusters in some cases. Was the clustering forced to give 4 clusters based on the assumption that this was the optimal number, or does the analysis suggest that? Please specify.
RESPONSE:
As the rule for the final number of clusters extracted from the analysis, the highest change of the similarity index was used. The number of clusters was not prespecified a-priory and this parameter revealed four clusters, which were suggested by the clustering algorithm. To illustrate this issue more precisely the sentence in the method section “By cutting the final dendrogram at the highest change of the similarity index we identified four morphometric patterns, consisting of the brain regions identified by the PCA (Figure 1).” was changed to “To define the number of relevant morphometric patterns, the dendrogram was cut at the highest change of the similarity index, which represents a measure of congruency of morphometric parameters among patients. This procedure revealed four morphometric patterns, consisting of the brain regions identified by the PCA (Figure 1).” (page 5, line 224-228).
5) There is no reference about the Control group in the Results about thickness nor in the cognitive tests. It is stated in the Methods section that they performed well in the cognitive test battery, but some reference in the statistics shown in Figure 3 is missing.
RESPONSE:
To account for deviations of morphometric parameters as well as clinical parameters from control groups and to make them comparable among centers, all values have been z-transformed. For morphometric parameters z-transformation was achieved by including a total of 348 healthy individuals (n=214 females, 61.5%) at the scanner sites. Z-transformation of clinical parameters was achieved by control data sets provided by the clinical test sets. The CERAD test battery is validated, standardized and normed on stratified age (range 49–92 years), education (range 7–20 years), and sex (German-speaking norm population: n = 1,100). (Berres, M.; Monsch, A.U.; Bernasconi, F.; Thalmann, B.; Stähelin, H.B. Normal ranges of neuropsychological tests for the diagnosis of Alzheimer’s disease. Stud. Health. Technol. Inform. 2000, 77, 195–199.).
6) Figure 3 is too small to be readable and becomes pixelated when zoomed in. Please correct it.
RESPONSE:
We now provided a new figure in in the JPG format in high resolution (2207 x 1546 pixels width/height). We assume that the low resolution of Figure 3 might also arise from converting into the PDF format.
Reviewer 2 Report
Lenhart et al. have prepared a paper on the automation of hippocampal imaging in Alzheimer's Disease. The paper is interesting, well-written, methodologically sound, and properly referenced. The introduction is adequate and presents the background and scope of the paper. The Materials and methods section is consistent and addresses all aspects detailed in the manuscript bulk. The results and discussions are thorough and supported by the necessary images and tables. Finally, the conclusions are reasonable and the strengths and limitations of the study are also reviewed. However, the following aspects might bring additional improvements to the manuscript:
- in Figure 1, the labels in the header should are somewhat unclear - perhaps adding "hippocampal sparing" to the third row would increase clarity or perhaps just choosing the same order/sequence of the four areas throughout the manuscript as to not confuse h-t-p with p-t involvement; also, the highlighted areas should be labeled and properly identified in the figure caption.
- please make sure that all abbreviations are explained in the text but also in the tables/figures - in some cases, some of them are missing from the captions (Table 2 - MMSE, maybe SD as well, if deemed appropriate since IQR is detailed).
- there is an inconsistency in the back matter, as there is a reference to funding in the acknowledgements tab, yet it is specified that no external funding was received in the funding tab - please clarify.
- some grammar errors throughout the text, some of which impact the readability (e.g. line 304) should be addressed.
Author Response
1) In Figure 1, the labels in the header shown are somewhat unclear - perhaps adding "hippocampal sparing" to the third row would increase clarity or perhaps just choosing the same order/sequence of the four areas throughout the manuscript as to not confuse h-t-p with p-t involvement; also, the highlighted areas should be labeled and properly identified in the figure caption.
RESPONSE:
Thank you for these suggestions. Please see also response 3 from reviewer 1. We modified the figure accordingly.
2) Please make sure that all abbreviations are explained in the text but also in the tables/figures - in some cases, some of them are missing from the captions (Table 2 - MMSE, maybe SD as well, if deemed appropriate since IQR is detailed).
RESPONSE:
Thank you for this comment. We added the missing abbreviations in Table 2.
MMSE, Mini-Mental State Examination
SD, Standard deviation
3) There is an inconsistency in the back matter, as there is a reference to funding in the acknowledgements tab, yet it is specified that no external funding was received in the funding tab - please clarify.
RESPONSE:
We did not receive any external funding, only the data itself were provided externally by the Austrian Alzheimer Society. To clarify that issue we suggest changing the wording on page 13, line 463/464 to: “The Austrian Alzheimer Society had no role in the study design, data analysis, decision to publish, or preparation of the manuscript.”
4) Some grammar errors throughout the text, some of which impact the readability (e.g. line 304) should be addressed.
RESPONSE:
Thank you for this comment. We checked the manuscript for further grammatical errors and made the recommended corrections on page 10, line 316.
Verbal fluency was most defective deteriorated in the hippocampal-temporal subgroup, and relatively preserved in the hippocampal predominant group when compared to the three other subgroups.